# Mexican Strains of *Anaplasma marginale*: A First Comparative Genomics and Phylogeographic Analysis

**DOI:** 10.3390/pathogens11080873

**Published:** 2022-08-02

**Authors:** Edgar Dantán-González, Rosa Estela Quiroz-Castañeda, Hugo Aguilar-Díaz, Itzel Amaro-Estrada, Fernando Martínez-Ocampo, Sergio Rodríguez-Camarillo

**Affiliations:** 1Laboratorio de Estudios Ecogenómicos, Centro de Investigación en Biotecnología, Universidad Autónoma del Estado de Morelos, Cuernavaca 62209, Mexico; edantan@uaem.mx (E.D.-G.); fernando.martinezo@uaem.mx (F.M.-O.); 2Centro Nacional de Investigación Disciplinaria en Salud Animal e Inocuidad, Instituto Nacional de Investigaciones Forestales, Agrícolas y Pecuarias, Jiutepec 62574, Mexico; aguilar.hugo@inifap.gob.mx (H.A.-D.); amaro.estrada@gmail.com (I.A.-E.); rodriguez.sergio@inifap.gob.mx (S.R.-C.)

**Keywords:** phylogeographic, bovine anaplasmosis, genomics, Mexican *Anaplasma marginale* strains

## Abstract

The One Health approach looks after animal welfare and demands constant monitoring of the strains that circulate globally to prevent outbreaks. Anaplasma marginale is the etiologic agent of bovine anaplasmosis and is endemic worldwide. This study aimed to analyze, for the first time, the genetic diversity of seven Mexican strains of *A. marginale* and their relationship with other strains reported. The main features of *A. marginale* were obtained by characterizing all 24 genomes reported so far. Genetic diversity and phylogeography were analyzed by characterizing the *msp1a* gene and 5′-UTR microsatellite sequences and constructing a phylogenetic tree with 540 concatenated genes of the core genome. The Mexican strains show 15 different repeat sequences in six MSP1a structures and have phylogeographic relationships with strains from North America, South America, and Asia, which confirms they are highly variable. Based on our results, we encourage the performance of genome sequencing of *A. marginale* strains to obtain a high assembly level of molecular markers and the performance of extensive phylogeographic analysis. Undoubtedly, genomic surveillance helps build a picture of how a pathogen changes and evolves in geographical regions. However, we cannot discard the study of relationships pathogens establish with ticks and how they have co-evolved to establish themselves as a successful transmission system.

## 1. Introduction

*Anaplasma marginale* is the causal agent of infectious and non-contagious anaplasmosis, characterized by jaundice, fever, anemia, abortion in pregnant cows, decreased milk production, weight loss, and, in severe cases, even death [1,2,3]. Bovine anaplasmosis has been reported in endemic regions in North, Central, and South America, Africa, Asia, and Australia and is considered to be an impediment to animal health and production [4,5]. Consequently, the productivity losses and economic costs associated with anaplasmosis are significant and have been estimated at USD 300–800 million [6]. Additionally, the genetic variability of different geographical *A. marginale* strains that have been reported worldwide creates difficulties in relation to controlling the disease [7]. The development of vaccines as a prevention strategy remains elusive, as only in some countries is the use of live vaccines of *Anaplasma centrale* allowed [8]. Veterinary genomics studies are a relatively recent field of interest with many techniques and approaches used to decipher the genomic content of pathogens of veterinary interest [9,10,11]. In Mexico, as in some other countries, knowledge of the genomic and genetic diversity of geographically different strains of *A. marginale* has been studied [12,13]. Whole-genome sequencing of *Anaplasma* strains using next-generation sequencing (NGS) technologies provides significant data collection with potential applications [14,15]. In addition, comparative genomics has emerged as an invaluable approach for illuminating evolutionary mechanisms and forces, as well as discovering the similarities and differences between genomes that may contribute to the development of a vaccine against the different strains that cause animal diseases [16,17].

With the idea of preserving animal health, the study of pathogens should not be studied in isolation, but rather as part of a combination where animal, human, and environmental health converge. Studying animal diseases, such as bovine anaplasmosis, provides an opportunity for enhanced understanding of the pathogens (genetic variants, genomic modifications, pathogenesis, virulence factors, etc.) concerning outbreak prevention by proposing alternatives to control diseases. Therefore, the study of pathogens that affect animal health should have a perspective based on a combination of genomics approaches and epidemiological and virulence studies that leads to proposed prevention and control strategies [18].

In this work, we characterized 24 genomes of *A. marginale* reported in the GenBank database and performed a genetic diversity and phylogeographic analysis which allowed for the characterization of *msp1a* gene and 5′-UTR microsatellite sequence construction of a phylogenetic tree with 540 concatenated genes belonging to the core genome. Our results suggest that the *A. marginale* genomes are highly conserved but differ in some regions. We also identified that the seven draft genomes of Mexican strains of *A. marginale* have a high assembly level with 32 to 46 contigs, a genome size of ~1.17 Mb, and G + C content of ~49.79%. Two draft genomes of Mexican strains have the same MSP1a repeats structure as previously reported sequences, but five draft genomes differ in the MSP1a repeats structure compared to previously reported sequences. The seven Mexican strains show 15 different sequence repeats in six different MSP1a structures and have phylogeographic relationships with strains from North America, South America, and Asia.

This work is the first phylogenetic analysis performed with the core genome of *A. marginale*, which represents a significant opportunity to elucidate their genetic diversity and the identification of proteins with potential applications in the control of bovine anaplasmosis.

## 2. Results

### 2.1. General Features of Anaplasma marginale Genomes

The significant genomic features of *A. marginale* are shown in Table 1. The GenBank database has reported 24 genomes of *A. marginale*: 2 complete genomes assembled in a single chromosome, 4 closed genomes assembled in a single incomplete chromosome (that contains 0.10–7.27% of unidentified nucleotides), and 18 draft genomes assembled in contigs or scaffolds. Table 1 shows that the three draft genomes of *A. marginale* strains Okeechobee, South Idaho, and Washington Okanogan have a low assembly level, with more than 300 contigs, larger genome size, and lower G + C content. In comparison, the seven draft genomes of Mexican strains were assembled from 32 to 46 contigs with a genome size of ~1.17 Mb and G + C content of ~49.79% (Table 1).

The automatic annotation of rRNA genes showed that all 24 genomes of *A. marginale* contain one copy of 23S, 5S, and 16S genes with lengths of ~2784, 114, and ~1491 bp, respectively (Appendix A). The mapping of rRNA genes shows that all 24 genomes contain an internal transcribed spacer (ITS) 80 bp distant from 23S and 5S rRNA genes (Appendix A). However, only two complete and four closed genomes contain a distance from 162,640 to 163,401 bp between the 23S-5S and 16S rRNA genes (Appendix A). The automatic annotation of tRNA genes showed that 19 genomes of *A. marginale*, including the seven Mexican strains, contain 37 tRNA genes (Table 1). The 19 genomes contain the same 37 tRNA genes representing 35 different codons of all 20 amino acids (Appendix A).

We detected the number of copies of the *msp1a* gene to determine if the seven draft genomes of Mexican strains contained genomic information of two or more different strains of *A. marginale* (possible superinfection) in the same genome. As a result, it was discovered that the seven draft genomes contain the complete sequence of a copy of the *msp1a* gene in longer length contig 1 (Appendix A). Thus, the seven draft genomes do not show a superinfection. Additionally, we obtained the tandem repeat sequences of MSP1a for seven Mexican strains using the Repeat Analyzer program (Table 2). Only two Mexican strains (MEX-01-001-01 and MEX-30-193-01) have the same repeats structure as previously reported sequences. Four strains (MEX-15-099-01, MEX-17-017-01, MEX-30-184-02, and MEX-31-096-01) differ in terms of the length and type of repeats structure in comparison to previously reported sequences (Table 2). The repeats structure of these four Mexican strains was verified to discard errors in the assembly of the *msp1a* genes (Appendix A).

As observed, the strain MEX-01-001-01 contains repeat structure 9 that has only been reported in Mexican strains (geographic regions of North America). The strain MEX-14-010-01 contains the repeat structure τ 22-2 13 18, which is identical to structures reported in Argentine strains (geographic regions of South America). The strains MEX-15-099-01 and MEX-30-193-01 contain repeats a, b, and G, which have been reported in strains from Argentina, Brazil, and Venezuela (geographic regions of South America). The strain MEX-17-017-01 contains repeat structure 14 that has also been reported in strains from China and the Philippines (geographic regions of Asia). Ultimately, the strains MEX-30-184-02 and MEX-31-096-01 contain repeats T and C, which have been reported in strains from the United States and Mexico (geographic regions of North America).

The results of the *msp1a* microsatellite sequences show that four Mexican strains (MEX-01-001-01, MEX-14-010-01, MEX-15-099-01, and MEX-30-193-01) have a microsatellite structure with *m* = 2, *n* = 7, and SD-ATG distance = 23 (Table 2). MSP1a first (RI) repeats 4 (MEX-01-001-01) and *t* (MEX-14-010-01) are unique sequences of ecoregion 1, and RI repeats a (MEX-15-099-01 and MEX-30-193-01) is reported in strains of ecoregion 1 but absent in strains of ecoregion 3 (Appendix A). MSP1a last (RL) repeat 9 (MEX-01-001-01) is a unique sequence for ecoregion 1, and RL repeats 18 (MEX-14-010-01) and G (MEX-15-099-01 and MEX-30-193-01) are reported in strains of ecoregions 1 and 3 (Appendix A).

These data suggest that the four Mexican strains are associated with ecoregion 1. Moreover, Table 2 shows that three Mexican strains (MEX-17-017-01, MEX-30-184-02, and MEX-31-096-01) have a microsatellite structure with *m* = 3, *n* = 5, and SD-ATG distance = 23. Thus, these three strains belong to genotype G, which is present with a frequency of 0.15, 0.14, 0.56, and 0.14 in ecoregions 1, 2, 3, and 4, respectively.

### 2.2. Phylogenetic and Pan-Genomic Analysis

The phylogenetic tree is based on 540 concatenated genes that belong to the core genome of 24 genomes of *A. marginale* (Figure 1). The model of nucleotide substitution was GTR + I + G. The phylogenetic tree shows that the strains from Puerto Rico and North America are grouped in a clade, evolutionarily distant from the Australian, Brazilian, and Mexican strains of *A. marginale.* The Mexican strains MEX-30-184-02 (Tlapacoyan, Veracruz) and MEX-31-096-01 (Tizimín, Yucatan) are closely related and are in a phylogenetic position close to the North American strains (Figure 1); the strains MEX-15-099-01 (Texcoco, Estado de Mexico) and MEX-30-193-01 (Tlapacoyan, Veracruz) are closely related, but these strains have different MSP1a repeat structures. Analysis of MSP1a and phylogenetic relationships suggest that three Mexican strains (MEX-01-001-01, MEX-30-184-02, and MEX-31-096-01) are related to strains from the geographic region of North America; three Mexican strains (MEX-14-010-01, MEX-15-099-01, and MEX-30-193-01) are related to strains from the geographic region of South America; and the strain MEX-17-017-01 is related to strains from the geographic region of Asia.

Pan-genomic analysis among the seven Mexican strains of *A. marginale* shows that the core genome comprises 883 CDS and 212 CDS for a single genome (Figure 2). Moreover, pan-genomic analysis among the 24 *A. marginale* genomes shows that the core genome comprises 534 CDS and 1155 CDS for a single genome (Figure 3).

The pan-genomic analysis among the 24 genomes of *A. marginale* shows that the single CDS and the core genome decrease significantly with respect to the pan-genomic analysis among the 7 genomes of Mexican strains. The Mexican strains contain 349 core genome CDS not shared with American, Australian, Brazilian, and Puerto Rican strains of *A. marginale*. Furthermore, the data suggest that some CDS are conserved in the Mexican strains while not present in the others, despite the close phylogenetic relationships.

### 2.3. Genome Comparison

A visual evaluation of the level of conserved genomic sequences showed that the draft genomes of seven Mexican strains of *A. marginale* are highly conserved, with almost 100% identity and almost complete coverage (Figure 4). The genes of rRNA, tRNA, and the type IV secretion system are conserved and show no significant differences. However, the family of MSP genes shows high diversity and/or variability (faded and faint colors) in all genomes. As additional information, the four closed genomes assembled in a single incomplete chromosome of strains Jaboticabal, Palmeira, Dawn, and Gypsy Plains show sequences gaps of unidentified nucleotides (Appendix A and Figure 4).

## 3. Discussion

Cattle health is the outcome of the balance between the microorganisms that inhabit them, either beneficial or pathogenic [20]. In this regard, how do we face the fact that animals for commercial purposes are mobilized while also considering the potential pathogens they may have? A starting point could be knowing the genetic diversity of *A. marginale*, with the idea of identifying which strains circulate in the cattle.

Mexico presents a wide genetic diversity of *A. marginale* strains related to strains from other continents [7]; this is probably due to the wide variety of ecological regions and the strategic commercial location that has allowed the movement of cattle and ticks (vectors) from different countries [21]. Ticks are part of their microbiota and possess a wide variety of microorganisms with different functions, many of them being potential pathogens to animals, including *A. marginale*, *Borrelia* spp., *Mycoplasma* spp., and *Coxiella* spp. among others [22].

The genomic sequencing of *A. marginale* strains during recent years has reported 24 genomes (2 complete genomes, 4 closed genomes, and 18 draft genomes) with 1.13 to 1.40 Mb of total length and G + C content from 46.73 to 49.84%.

Specifically, the comparison of the 18 draft genomes of *A. marginale* strains from the US, Puerto Rico, and Mexico suggests that the 7 draft genomes from Mexico have a high assembly level with a lower number of contigs than the 11 draft genomes of strains from Puerto Rico (59 contigs) and the United States (57–403 contigs and 44 scaffolds). Moreover, the 7 draft genomes of Mexican strains contain from 1178 to 1218 genes and from 1138 to 1178 CDS (Table 1), a similar number to 14 genomes of *A. marginale* from other countries and a smaller number compared to the 3 draft genomes with a low assembly level. These data suggest that a low rate of gain/loss of genes occurred through evolution in the *A. marginale* genomes.

Additionally, the annotation of rRNA genes suggested that all 24 genomes of *A. marginale* contain one copy of 23S, 5S, and 16S rRNA separated by a distance greater than 162 kb, and an ITS of 80 bp between 23S and 5S genes (Appendix A). As for the annotation of tRNA, the results showed that 19 genomes of *A. marginale* contain 37 tRNA genes for 35 different codons of all 20 aa, one copy for each tRNA gene, except the tRNA-Met gene which contained three copies, maybe due to the importance of initiating protein synthesis.

On the other hand, in Mexico and worldwide, identification and molecular characterization of the genetic diversity of *A. marginale* isolates are based on markers such as the variable region of MSP1a. The results obtained in this work from MSP1a suggest that the Mexican strains MEX-30-184-02 and MEX-31-096-01 are associated with ecoregion 1, while strain MEX-17-017-01 was not associated with any of the four ecoregions reported by [19]. Additionally, all Mexican strains, except MEX-17-017-01, are transmitted by ticks associated with ecoregion 1. Ecoregion 1 mainly extends over large areas of central Africa and central South America, primarily Argentina and southern Brazil. It is involved in a region with medium to high Normalized Difference Vegetation Index (NDVI) values, i.e., shrub and grassland to temperate and tropical rainforests. This region has the highest recorded temperature and 1000 mm of annual rainfall [19]. Thus, the environmental factors are dynamic and could change daily. It is essential to highlight that obtention of the complete sequences of *msp1a* genes by several molecular methods increases the error rate in the consensus sequences of these genes.

These results suggest that a percentage of *msp1a* gene sequences, obtained by several steps of molecular methods and reported in databases, may contain errors affecting the analyses of genetic diversity or geographic distribution. It could not be excluded that, during massive sequencing, errors might occur, especially in assemblies with low quality or that are still drafts. Therefore, the variable region sequence of *msp1a* genes could be confirmed or corrected by genome sequencing of *A. marginale* strains.

Moreover, genome sequencing provides complete sequences with a higher assembly level of *msp1a* genes. Surprisingly, we detected some draft genomes of Mexican strains of *A. marginale* that contain MSP1a repeat structures that differs from previously reported sequences. This fact highlights the importance of genome sequencing [7]. Currently, there are few reports of *msp1a* gene sequences by genome sequencing, while the *msp1a* sequences amplified by PCR are more abundant and include end-point, nested, and semi-nested PCR [23,24,25] In this regard, genome sequencing is proposed as an alternative to PCR amplification for studying the genetic diversity of *A. marginale*. At this point, we must emphasize that short-read sequencing methods, such as the one used in this work, allow high-quality reads assembled into genomes to be obtained. However, the obtained genomes are still imperfect and may contain assembly errors, which can mask interesting signals and propagate into false identification of candidate genes and inaccurate gene annotation [26]. Therefore, long-read sequencing (LRS) technologies emerge to improve genome assembly quality, and may offer refinements in the characterization of genetic variation and regions that are difficult to assess with prevailing NGS approaches [27].

Genome sequencing of the *msp1a* microsatellite regions of Mexican strains of *A. marginale* revealed that MEX-01-001-01, MEX-14-010-01, MEX-15-099-01, and MEX-30-193-01 belong to genotype E as proposed by Estrada-Peña et al. [20], which is present with a frequency of 0.75 in ecoregion 1. Thus, these four Mexican strains are associated with ecoregion 1. As for Mexican strains MEX-17-017-01, MEX-30-184-02, and MEX-31-096-01, their microsatellite structure showed that they belong to genotype G, which is present with a frequency of 0.15, 0.14, 0.56, and 0.14 in ecoregions 1-4, respectively. Interestingly, MSP1a RI repeats 12 and 14 (MEX-17-017-01) have not been reported in strains of the four ecoregions, while RI repeat T (MEX-30-184-02 and MEX-31-096-01) has been reported in ecoregions 1 and 2 and RL repeat C (MEX-30-184-02 and MEX-31-096-01) in ecoregions 1 and 4 (Appendix A).

This work is the first phylogenetic analysis performed with the core genome of *A. marginale* (540 concatenated genes). The phylogenetic reconstruction showed that Mexican *A. marginale* are related to strains from the geographic region of North America (MEX-01-001-01, MEX-30-184-02, and MEX-31-096-01), South America (MEX-14-010-01, MEX-15-099-01, and MEX-30-193-01), and Asia (MEX-17-017-01). However, it would be necessary to obtain *A. marginale* genome sequences from countries in Africa, Asia, and Europe to acquire a more defined tree topology (Figure 1). Additionally, the pan-genomic analysis showed that Mexican strains share specific local genomic and global information with strains from other continents.

Finally, comparative genomics revealed that *A. marginale* genomes are highly conserved but not identical, since the ANIm algorithm calculated alignment coverage and identity >93 and 98%, respectively.

As was observed in this study, the geographic strains of *A. marginale* are highly variable. This genetic variability is most likely due to cattle movement for commercial purposes and the presence of ticks that are transmitters of this pathogen. Additionally, the livestock trade has led to the emergence of strains of *A. marginale* from other countries, which has enriched the diversity that already existed. However, an essential factor to consider is the relationships pathogens establish with ticks and how they have co-evolved to establish themselves as a successful transmission system. Derived from our study, we propose a deep investigation of the interactions between ticks and *A. marginale* from the perspective of how populations of ticks impact *A. marginale* and vice-versa.

## 4. Materials and Methods

### 4.1. Sample Collection and Nomenclature

Blood samples were obtained from cattle infected in the field from different geographic regions of Mexico: Aguascalientes, Aguascalientes (MEX-01-001-01); Atitalaquia, Hidalgo (MEX-14-010-01); Texcoco, Estado de México (MEX-15-099-01); Puente de Ixtla, Morelos (MEX-17-017-01); Tlapacoyan, Veracruz (MEX-30-184-02); Veracruz, Veracruz (MEX-30-193-01); and Tizimín, Yucatán (MEX-31-096-01). The animals from which the blood was collected presented typical clinical signs of disease, and diagnosis of the disease was made using nested Polymerase Chain Reaction (PCR), amplifying the msp5 gene to check for the presence of *A. marginale*. The name of the isolates describes the state, municipality, and number of the isolate obtained for this study according to the information used by the Instituto Nacional de Estadística y Geografía (INEGI by acronym in Spanish).

### 4.2. Genome Sequences and Annotation

In Mexico, the Anaplasmosis Unit of the Instituto Nacional de Investigaciones Forestales, Agrícolas y Pecuarias (INIFAP by acronym in Spanish) reported the draft genomes of seven Mexican strains of *A. marginale* isolated from different geographic locations [12,13,28]. These seven genomes are the only ones reported for Mexican strains worldwide. Figure 5 shows the geographical regions from where the seven bovine blood samples infected with Mexican strains of *A. marginale* were obtained.

The 24 genomes of *A. marginale* strains from Australia, Brazil, Mexico, Puerto Rico, and the United States included in this study and reported to date in the GenBank (https://www.ncbi.nlm.nih.gov/genbank/, accessed on 21 January 2022) database are listed in Appendix A. The general features of the 24 genomes of *A. marginale* were obtained using the QUAST (Quality Assessment Tool for Genome Assemblies) (v5.0.2) program [29] with default settings. The 24 genomes were annotated automatically to predict the open reading frames (ORFs) and coding sequences (CDS) using the RAST (Rapid Annotation using Subsystem Technology) (v2.0) (http://rast.nmpdr.org/, accessed on 18 February 2022) server [30] with the Classic RAST algorithm. Additionally, the automatic annotation of ribosomal (rRNAs) and transfer (tRNAs) RNA genes was carried out using the RNAmmer (v1.2) (http://www.cbs.dtu.dk/services/RNAmmer/, accessed on 21 February 2022) [31] and ARAGORN (v1.2.38) (http://mbio-serv2.mbioekol.lu.se/ARAGORN/, accessed on 22 February 2022) [32] servers, respectively.

### 4.3. Analysis of MSP1a

We used the MSP1a protein of *A. marginale* strain St. Maries (GenBank accession number AAV86554.1) as a query sequence to identify the number of copies and location of *msp1a* genes in the seven draft genomes of Mexican strains of *A. marginale* using the tBlastn (https://blast.ncbi.nlm.nih.gov/Blast.cgi, accessed on 5 March 2022) program [33]. The *msp1a* gene sequences that had an alignment coverage and identity greater than 50 and 80%, respectively, were selected.

The tandem repeat sequences of the variable region of MSP1a proteins were obtained using the Repeat Analyzer (v2.8) program [34] with default settings. Additionally, the tandem repeat sequences were compared against the *A. marginale* database (updated 13 December 2017, with 412 strains and 274 repeat sequences) of the Repeat Analyzer (v2.8) [34] program to obtain the possible geographical distribution worldwide.

The *msp1a* 5′-UTR (Untranslated Region) microsatellite sequences were located between the putative Shine–Dalgarno (SD) sequence (GTAGG) and the translation initiation codon (ATG), (SD-ATG) [19]. The structure of microsatellites (bold) was determined by GTAGG (G/A TTT)m (GT)n T ATG [19]. The analysis of genotypes was performed according to the nomenclature proposed by de la Fuente et al. (2007) [35]. The SD-ATG distance was calculated in nucleotides according to the formula: (4 × m) + (2 × n) + 1 [19].

### 4.4. Phylogenetic and Pan-Genomic Analysis

For the phylogenetic reconstruction, the 24 genomes of *A. marginale* were annotated automatically using the Prokka (rapid prokaryotic genome annotation) (v1.12) program [36] with default settings. The GFF (General Feature Format) annotation files were used as input files by the Roary (the pan-genome pipeline) (v3.12.0) program [37], which made the multiple alignments between the core genomes using the option to create a multiFASTA alignment of core genes (-e). The jModelTest (v2.1.10) program [38] was used to select the best model of nucleotide substitution with the Akaike information criterion (AIC). The phylogenetic tree was estimated under the Maximum-Likelihood (ML) method using the PhyML (v3.1) program [39] with 1000 bootstrap replicates. The phylogenetic tree was visualized and edited using the FigTree (v1.4.4) (http://tree.bio.ed.ac.uk/software/figtree/, accessed on 6 March 2022) program.

Two pan-genomic analyses were performed using the GET_HOMOLOGUES (v11042019) software package [40] with the following options: (i) among the 7 genomes of Mexican strains; and (ii) among the 24 genomes of *A. marginale*. Briefly, the FAA (Fasta Amino Acid) annotation files of *A. marginale* strains that were obtained with the RAST server (see Section 4.2) were used as input files by the GET_HOMOLOGUES (v11042019) software package [40]. The get_homologues.pl and compare_clusters.pl Perl scripts were used to compute a consensus pan-genome, which resulted from the clustering of the all-against-all Blastp results with the COGtriangles and OMCL algorithms. The pan-genomic analyses were performed using the binary (presence–absence) matrix.

### 4.5. Comparative Genomics

The level of conserved genomic sequences was visualized by alignment of the seven genomes of Mexican strains against the reference genome of *A. marginale* strain St. Maries (GenBank accession number CP000030.1) and five genomes of *A. marginale* strains from Australia (Dawn and Gypsy Plains), Brazil (Jaboticabal and Palmeira), and the United States (complete genome of strain Florida) using the BRIG (BLAST Ring Image Generator) (v0.95) [41] software package with default settings. The circular comparative genomic map was constructed using GenBank files (gbk format) and the NCBI local blast-2.9.0 + suite. Ultimately, the average nucleotide identity (ANI) values were calculated by comparing the 13 genomes of *A. marginale* (above in this section) using the calculate_ani.py Python script (https://github.com/ctSkennerton/scriptShed/blob/master/calculate_ani.py, accessed on 10 March 2022) with the ANIm (by MUMmer) algorithm.

## 5. Conclusions

Currently, the potential represented by the massive sequencing of genomes is undeniable. Pathogens of veterinary importance, such as *A. marginale*, represent key targets. In this work, we focus on analyzing the phylogeographic relationships that exist in the strains of *A. marginale* reported worldwide and those that we have identified in Mexico.

We found that the structure of the MSP1a repeats of five Mexican strains did not coincide with those reported in the literature. Thus, we propose that these errors can decrease considerably through massive sequencing of the genomes and not only of the *msp1a* gene. On the other hand, the phylogenetic analysis of the core genome of 24 genomes of *A. marginale* showed that the Mexican strains are related to strains from regions of North America, South America, and Asia. Additionally, pan-genomic analysis between the seven Mexican strains revealed that the core genome contains 883 genes, while this value decreases to 534 genes when using the 24 genomes of *A. marginale*, showing that there are genes that only present in the national strains. It is worth mentioning that it is precisely these unique genes which are of interest because they represent potential targets for vaccine development to prevent anaplasmosis in our country, and we are currently working on this approach.

## Figures and Tables

**Figure 1 pathogens-11-00873-f001:**
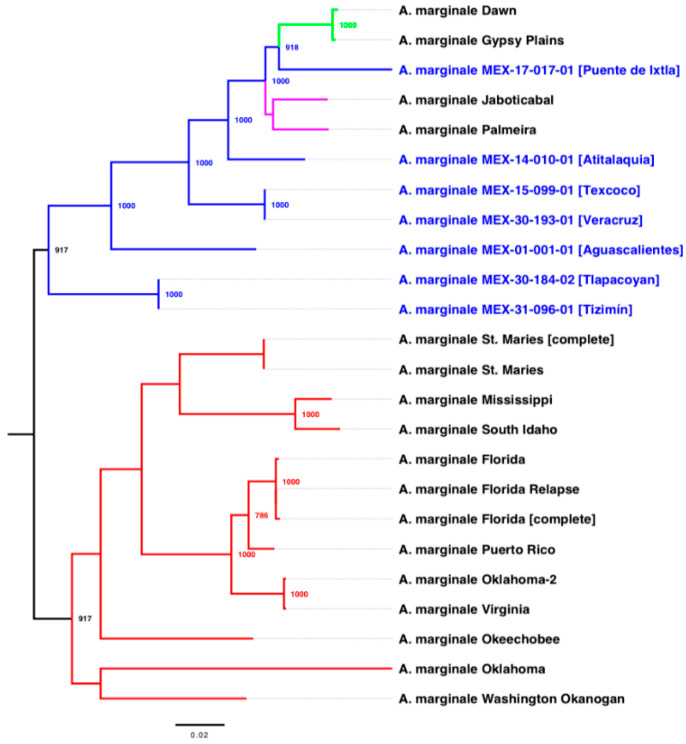
Phylogenetic relationships based on the core genome (540 concatenated genes) of seven Mexican strains (blue lines and letters), 12 genomes of North American strains and one strain from Puerto Rico (red lines), two Australian strains (green lines) and two Brazilian strains (pink lines) of *A. marginale*. The phylogenetic tree was obtained using the PhyML program with the maximum likelihood method and 1000 bootstrap replicates. Bootstrap values (>70%) are displayed in the nodes.

**Figure 2 pathogens-11-00873-f002:**
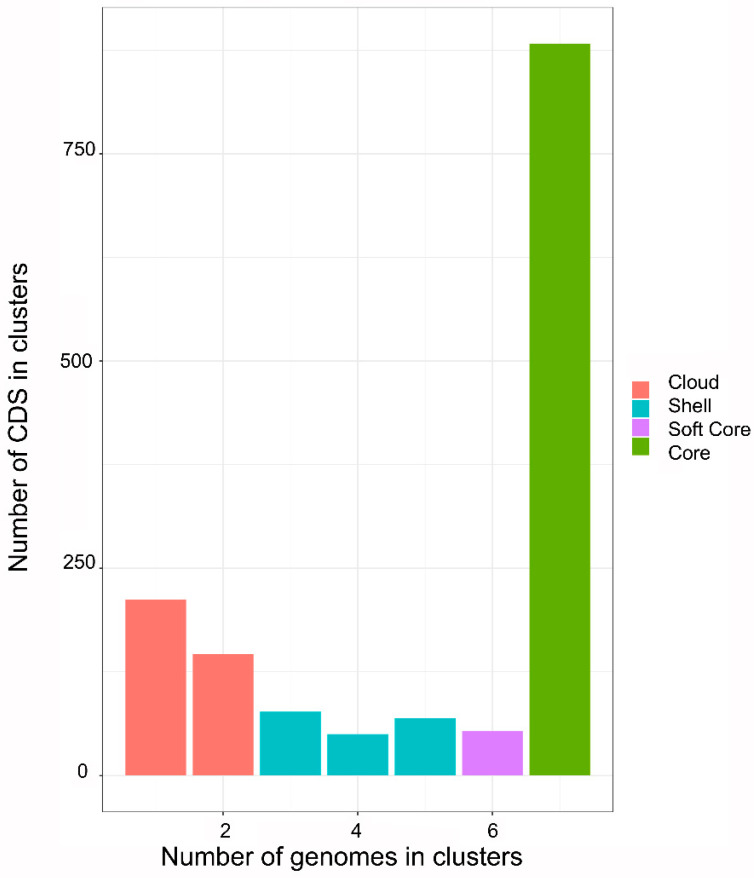
Pan-genome distribution in four categories (Cloud, Shell, Soft Core, and Core) of Mexican strains of *A. marginale*. The core genome contains 883 coding sequences (CDS) shared by the seven genomes. Cloud genome (red) containing between 0 and less than 15% of the total CDS; Shell (blue) containing between 15 and less than 95% of the total CDS; Soft Core (purple) containing between 95 and less than 100% of the total CDS; and Core (green) containing 100% of the total CDS.

**Figure 3 pathogens-11-00873-f003:**
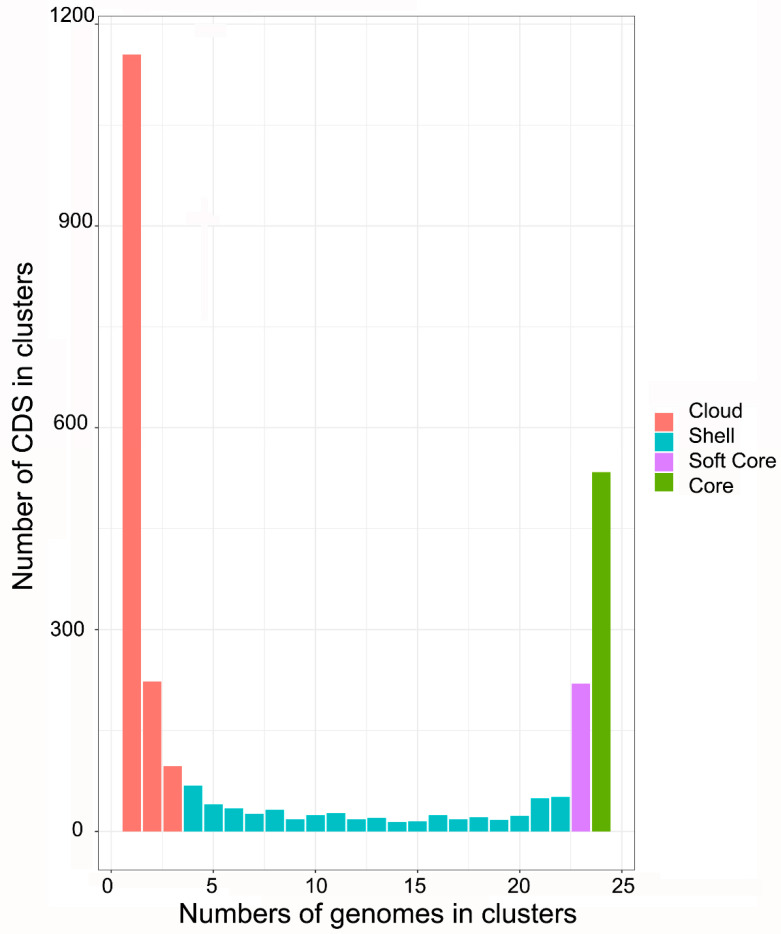
Pan-genome distribution in four categories (Cloud, Shell, Soft core, and Core) of coding sequences (CDS) shared by 24 *A. marginale* genomes. The core genome contains 534 CDS shared by the seven genomes. Cloud (red) containing between 0 and less than 15% of the total CDS; Shell (blue) containing between 15 and less than 95% of the total CDS; Soft Core (purple) containing between 95 and less than 100% of the total CDS; and Core (green) containing 100% of the total CDS.

**Figure 4 pathogens-11-00873-f004:**
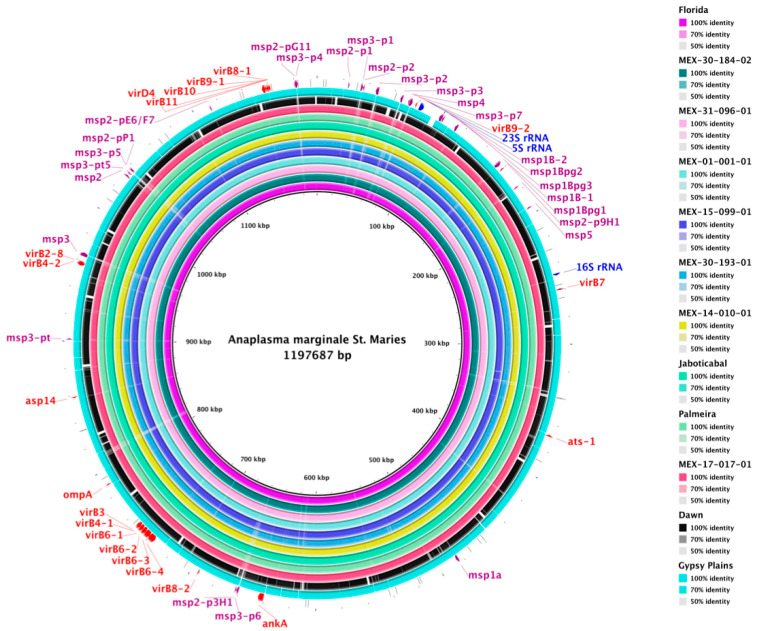
Whole-genome comparisons in *A. marginale*. From outer to inner ring: strains St. Maries (as a reference genome), Florida, MEX-30-184-02 (Tlapacoyan, Veracruz), MEX-31-096-01 (Tizimín, Yucatán), MEX-01-001-01 (Aguascalientes, Aguascalientes), MEX-15-099-01 (Texcoco, Estado de México), MEX-30-193-01 (Veracruz, Veracruz), MEX-14-010-01 (Atitalaquia, Hidalgo), Jaboticabal, Palmeira, MEX-17-017-01 (Puente de Ixtla, Morelos), Dawn, and Gypsy Plains. The outermost ring highlights the rRNA (blue arrows), tRNA (black lines), msp (purple arrows), and type IV secretion system (red arrows) genes. The color intensity in each ring represents the BLAST match identity: solid color shows 100% identity, faded color shows 70% identity, and faint color shows 50% identity according to BRIG output.

**Figure 5 pathogens-11-00873-f005:**
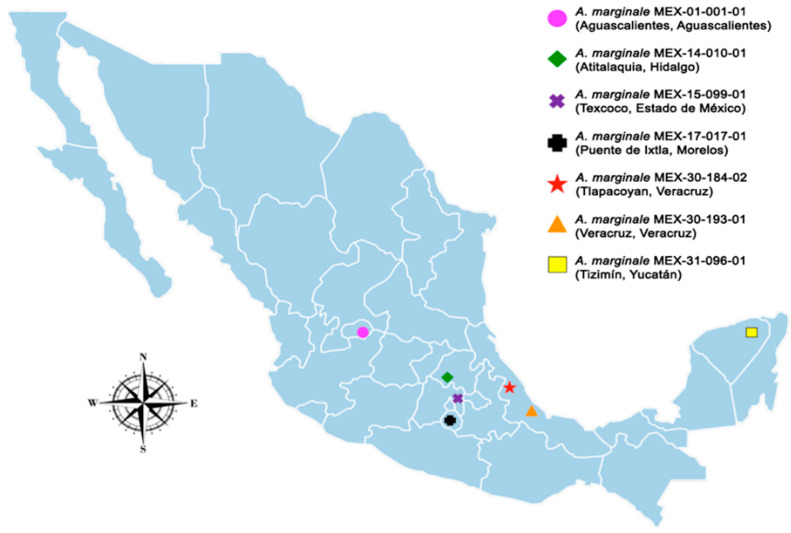
The political division of Mexico, showing states where the blood samples for genomic sequencing were sampled.

**Table 1 pathogens-11-00873-t001:** General features of 24 genomes of *Anaplasma marginale*.

Organism	Assembly Level	Total Length (bp) *	G + C Content (%) *	Genes **	CDS **	rRNAs ^#^	tRNAs ^##^
*A. marginale* Dawn	Incomplete chromosome	1,196,760	49.73	1145	1107	3	35
*A. marginale* Gypsy Plains	Incomplete chromosome	1,198,622	49.72	1189	1149	3	37
*A. marginale* Jaboticabal	Incomplete chromosome	1,195,321	49.77	1238	1198	3	37
*A. marginale* Palmeira	Incomplete chromosome	1,195,200	49.75	1219	1179	3	37
*A. marginale* MEX-01-001-01	Draft (34 contigs)	1,179,425	49.79	1218	1178	3	37
*A. marginale* MEX-14-010-01	Draft (46 contigs)	1,172,327	49.79	1190	1150	3	37
*A. marginale* MEX-15-099-01	Draft (32 contigs)	1,169,440	49.79	1185	1145	3	37
*A. marginale* MEX-17-017-01	Draft (41 contigs)	1,172,716	49.79	1203	1163	3	37
*A. marginale* MEX-30-184-02	Draft (40 contigs)	1,176,681	49.79	1205	1165	3	37
*A. marginale* MEX-30-193-01	Draft (41 contigs)	1,167,111	49.80	1178	1138	3	37
*A. marginale* MEX-31-096-01	Draft (43 contigs)	1,176,579	49.79	1204	1164	3	37
*A. marginale* Puerto Rico	Draft (59 contigs)	1,158,530	49.80	1220	1180	3	37
*A. marginale* Florida	Complete chromosome	1,202,435	49.77	1227	1187	3	37
*A. marginale* Florida	Draft (204 contigs)	1,136,981	49.84	1186	1147	3	35
*A. marginale* Florida Relapse	Draft (61 contigs)	1,154,411	49.81	1189	1149	3	37
*A. marginale* Mississippi	Draft (82 contigs)	1,141,520	49.79	1208	1168	3	37
*A. marginale* Okeechobee	Draft (403 contigs)	1,390,987	47.49	1267	1225	3	38
*A. marginale* Oklahoma	Draft (57 contigs)	1,156,921	49.82	1167	1127	3	37
*A. marginale* Oklahoma-2	Draft (44 scaffolds)	1,160,766	49.79	1188	1148	3	37
*A. marginale* South Idaho	Draft (358 contigs)	1,409,432	46.73	1316	1262	3	47
*A. marginale* St. Maries	Complete chromosome	1,197,687	49.76	1250	1210	3	37
*A. marginale* St. Maries	Draft (60 contigs)	1,155,236	49.79	1194	1154	3	37
*A. marginale* Virginia	Draft (70 contigs)	1,153,875	49.79	1241	1201	3	37
*A. marginale* Washington Okanogan	Draft (332 contigs)	1,383,255	46.94	1331	1274	3	52

CDS: Coding sequences. * Data obtained with the QUAST program. ** Data obtained with the RAST server. ^#^ Data obtained with the RNAmmer server. ^##^ Data obtained with the ARAGORN server.

**Table 2 pathogens-11-00873-t002:** Features of Mexican strains genomes of *Anaplasma marginale* based on geographic region and genotype frequency.

Organism	Geographic Region	Repeats Structure Previously Reported *	Repeats Structure Reported in This Work **	*m*	*n*	SD-ATG Distance	Genotype ***	Genotype Frequency Per Ecoregion Cluster ***			
								1	2	3	4
*A. marginale* MEX-01-001-01	Aguascalientes, Aguascalientes	4 9 10 11 9	4 9 10 11 9	2	7	23	E	0.75	0.00	0.25	0.00
*A. marginale* MEX-14-010-01	Atitalaquia, Hidalgo	τ 57 13 18	τ 22-2 13 18	2	7	23	E	0.75	0.00	0.25	0.00
*A. marginale* MEX-15-099-01	Texcoco, Estado de México	α β β Γ	α β Γ	2	7	23	E	0.75	0.00	0.25	0.00
*A. marginale* MEX-17-017-01	Puente de Ixtla, Morelos	12 13 14	12 14	3	5	23	G	0.15	0.14	0.56	0.14
*A. marginale* MEX-30-184-02	Tlapacoyan, Veracruz	73 β β β Γ	T C	3	5	23	G	0.15	0.14	0.56	0.14
*A. marginale* MEX-30-193-01	Veracruz, Veracruz	α β β Γ	α β β Γ	2	7	23	E	0.75	0.00	0.25	0.00
*A. marginale* MEX-31-096-01	Tizimín, Yucatán	T C B B C B π	T C	3	5	23	G	0.15	0.14	0.56	0.14

* Data reported by Jiménez-Ocampo et al. [7]. ** Data obtained with the Repeat Analyzer program. *** Data obtained by Estrada-Peña et al. [19].

## Data Availability

The data presented in this study are available on request from the corresponding author. The data are not publicly available due to the on-course research investigation.

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
