# Peer review of "Mexican Strains of Anaplasma marginale: A First Comparative Genomics and Phylogeographic Analysis"

_pathogens, 2022, doi:10.3390/pathogens11080873_

Round 1

Reviewer 1 Report

1) The original publication developing the PCR of Msp1a typing was not cited by the authors and this should be rectified and the manuscript updated:

A msp1α polymerase chain reaction assay for specific detection and differentiation of Anaplasma marginale isolates - ScienceDirect

2) the discussion could be improved, there are some short paragraphs and references to figures in results - which should be done in a discussion section. The authors could have discussed how to improve genome assemblies noting the data so far has been obtained through short read sequencing, long read sequencing may improve the assembles better than the the current available.

3) when referring to species generally, please use spp. not just the genus name eg. Borrelia, Coxiella etc..

Author Response

We appreciate the effort and dedication of the reviewers toward the submitted manuscript. We know this revision activity significantly enriches and improves our work, which is why we appreciate the interest and all the comments expressed. In this new version of the manuscript, we have addressed the comments and suggestions of each reviewer.

Response to Reviewer 1 

The authors appreciate all the comments that have improved the manuscript. The reviewer´s comments definitively have contributed to have a better version of this paper, therefore, we thank for your review and observations. 

Comments and Suggestions for Authors

1) The original publication developing the PCR of Msp1a typing was not cited by the authors and this should be rectified and the manuscript updated:

A msp1α polymerase chain reaction assay for specific detection and differentiation of Anaplasma marginale isolates - ScienceDirect

  1. The authors thank the valuable comment. We have already included the reference Lew et al., 2002 in Discussion. (Line 375).

2) the discussion could be improved, there are some short paragraphs and references to figures in results - which should be done in a discussion section. The authors could have discussed how to improve genome assemblies noting the data so far has been obtained through short read sequencing, long read sequencing may improve the assembles better than the the current available.

  1. Thank you for your valuable comment. The discussion has been enriched with information presented in the results section. The authors integrated the additional information to achieve a better interpretation of the results and their discussion. In this new version, the discussion has been improved as suggested by the reviewer. (Lines 319-415).

3) when referring to species generally, please use spp. not just the genus name eg. Borrelia, Coxiella etc..

  1. The authors appreciate the comment. We have already corrected the names of the bacteria as the reviewer suggested. (Line 329).

Reviewer 2 Report

The manuscript by Dantán-González et al. compares the genome of several Anaplasma marginale strains isolated in Mexico. The manuscript is well-written. The introduction provides relevant information for the reader understanding of the context of the study and the methods are sounds. This topic is important and genome sequencing studies of tick-borne pathogens are underrepresented, compared to analysis performed with single genes. However, considering that the genome sequencing of these strains was already reported (Microbiol Resour Announc. 2019;8(45):e01184-19; Int J Genomics. 2020;2020:5902029, and  Microbiol Resour Announc. 2018;7(16):e01101-18.) one would expect more in-depth comparisons here. Some recommendations are include below.

 Abstract

A conclusion statement is missing in the abstract

Results

Line 122: Not clear why the authors placed so much weight to the analysis of msp1a disconnected from the genome context. In this paper, the msp1a analysis appears as a stand-alone section. Instead, this gene could be analyzed in the context of genome comparison (Line 237) and together with other MSP for which the authors mentioned high variation was found (line 242). My point, now that the full genomes are available, why detail only the analysis of msp1a.

Lines 94-102: One would expect more details about the functional categories of the genes that were differentially present between the different Mexican strains, and between the Mexican strains and other strains with available genomes. A Venn diagram could help representing the differences in gen content. An additional supplementary table listing the differentially present genes could also be helpful.

Line 237: Change to ‘Genome comparison’. Also, considering that differences in virulence for some of the strains reported here (Microbiol Resour Announc. 2019;8(45):e01184-19; Int J Genomics. 2020;2020:5902029, and  Microbiol Resour Announc. 2018;7(16):e01101-18.) and those available in GenBank are known, can the authors performed some additional analysis to reveal genetic traits of A. marginale associated with virulence?

Methods

General

Describe the origin of the Mexican strains in detail. This is needed, as even the papers reporting these strains for the first time (Microbiol Resour Announc. 2019;8(45):e01184-19; Int J Genomics. 2020;2020:5902029, and  Microbiol Resour Announc. 2018;7(16):e01101-18.) do not include such description. For example

Figures

Figure 2 It is not clear what the authors are showing in this figure. Please, make clearer. Same for figure 3.

Conclusions

Conclusions are missing in the manuscript

Author Response

We appreciate the effort and dedication of the reviewers toward the submitted manuscript. We know this revision activity significantly enriches and improves our work, which is why we appreciate the interest and all the comments expressed. In this new version of the manuscript, we have addressed the comments and suggestions of each reviewer. Please see the attachment that contains the corrected manuscript.

Response to Reviewer 2

  1. The authors appreciate the reviewer's comments that enrich and improve the quality of the manuscript. Suggestions and modifications were made and responses to comments are presented below.

Abstract

A conclusion statement is missing in the abstract

  1. The authors appreciate the observation. In the abstract, a concluding sentence has been added and the information in the abstract has been revised and corrected again. (Lines 29-43)

Results

Line 122: Not clear why the authors placed so much weight to the analysis of msp1a disconnected from the genome context. In this paper, the msp1a analysis appears as a stand-alone section. Instead, this gene could be analyzed in the context of genome comparison (Line 237) and together with other MSP for which the authors mentioned high variation was found (line 242). My point, now that the full genomes are available, why detail only the analysis of msp1a.

  1. The authors greatly appreciate your observation. We should mention that the analysis focused on the msp1a gene because it is the most widely used molecular marker for analyzing the genetic diversity among A. marginale strains. By focusing on this marker, the authors intend to emphasize the genetic differences between A. marginale strains with different geographic locations to elucidate the phylogeographic relationship between all strains reported worldwide. The authors do not rule out the importance of analyzing the rest of the genes in greater depth, especially those related to virulence. However, that topic is being worked on in a manuscript for further publication.

Lines 94-102: One would expect more details about the functional categories of the genes that were differentially present between the different Mexican strains, and between the Mexican strains and other strains with available genomes. A Venn diagram could help representing the differences in gen content. An additional supplementary table listing the differentially present genes could also be helpful.

  1. Thank you very much for your valuable comment. In this regard, we consider significant to mention that the information about the genes differentially present in the Mexican strains and the other strains with available genomes is being saved for publication in a manuscript we are currently writing. Sharing this information at this time would significantly compromise the manuscript we are currently working on. Therefore, we cannot provide this information at this moment.

Line 237: Change to ‘Genome comparison’. Also, considering that differences in virulence for some of the strains reported here (Microbiol Resour Announc. 2019;8(45):e01184-19; Int J Genomics. 2020;2020:5902029, and  Microbiol Resour Announc. 2018;7(16):e01101-18.) and those available in GenBank are known, can the authors performed some additional analysis to reveal genetic traits of A. marginale associated with virulence?

  1. The authors appreciate the collaboration of the reviewer and the vision he has to propose additional analyses. The change to “comparative genomics” to “Genome comparison” was done. (Line 232). As we mentioned above, we are currently working on a manuscript that precisely contains the information requested by the reviewer. It would be a great pleasure for us to be able to share it in this manuscript, but we are unable to do so without compromising the manuscript that we are currently developing.

Methods

General

Describe the origin of the Mexican strains in detail. This is needed, as even the papers reporting these strains for the first time (Microbiol Resour Announc. 2019;8(45):e01184-19; Int J Genomics. 2020;2020:5902029, and  Microbiol Resour Announc. 2018;7(16):e01101-18.) do not include such description. For example

  1. Thank you for the observation. As the reviewer mention, this information was not included in the original manuscript. Thus, the authors included the correct information about the origin of the Mexican strains in Materials and Methods section: (Lines 417-428).

“Sample collection and nomenclature

Blood samples were obtained from cattle infected in the field, from different geographic regions of Mexico: Aguascalientes, Aguascalientes (MEX-01-001-01); Atitalaquia, Hidalgo (MEX-14-010-01); Texcoco, Estado de México (MEX-15-099-01); Puente de Ixtla, Morelos (MEX-17-017-01); Tlapacoyan, Veracruz (MEX-30-184-02); Veracruz, Veracruz (MEX-30-193-01); and Ti-zimín, Yucatán (MEX-31-096-01). The animals from which the blood was collected presented typical clinical signs of the disease, and diagnosis of the disease was made using nested Polymerase Chain Reaction (PCR), amplifying the msp5 gene to check for the presence of A. marginale. The name of the isolates describes the state, municipality, and number of the isolate obtained for this study according to the information used by Instituto Nacional de de Estadística y Geografía (INEGI by acronym in Spanish).

Figures

Figure 2 It is not clear what the authors are showing in this figure. Please, make clearer. Same for figure 3.

  1. The authors value and appreciate the observation of the reviewer. On figures 2 and 3, we have included missing information that had not been included in the first version of the manuscript, that is, information on the x and y axes in the graphs and a better explanation in the figure caption and in the text. (Lines 196-231).

Conclusions

Conclusions are missing in the manuscript

  1. Thank you for the observation. The authors included section 5. Conclusions in the new version of the manuscript. (Lines 510-526).

Round 2

Reviewer 2 Report

The reviewer thank the authors for addressing the raised points.